# The Clinical Pulmonary Infection Score Combined with Procalcitonin and Lung Ultrasound (CPIS-PLUS), a Good Tool for Ventilator Associated Pneumonia Early Diagnosis in Pediatrics

**DOI:** 10.3390/children11050592

**Published:** 2024-05-14

**Authors:** Judit Becerra-Hervás, Carmina Guitart, Aina Covas, Sara Bobillo-Pérez, Javier Rodríguez-Fanjul, Josep L. Carrasco-Jordan, Francisco José Cambra Lasaosa, Iolanda Jordan, Mònica Balaguer

**Affiliations:** 1Pediatric Intensive Care Unit, Hospital Sant Joan de Déu, University of Barcelona, Passeig de Sant Joan de Déu, 2, 08950 Barcelona, Spain; judit.becerra@sjd.es (J.B.-H.); carmina.guitart@alderhey.nhs.uk (C.G.); sara.bobillo@sjd.es (S.B.-P.); franciscojose.cambra@sjd.es (F.J.C.L.); monica.balaguer@sjd.es (M.B.); 2Faculty of Medicine, University of Barcelona, c. Casanova, 143, 08036 Barcelona, Spain; jlcarrasco@ub.edu; 3Immunological and Respiratory Disorders in the Paediatric Critical Patient Research Group, Institut de Recerca Sant Joan de Déu, University of Barcelona, 08950 Barcelona, Spain; 4Neonatal Intensive Care Unit, Hospital Sant Joan de Déu, University of Barcelona, 08950 Barcelona, Spain; aina.covas@sjd.es; 5Neonatal Intensive Care Unit, Department of Paediatrics, Hospital Germans Trias i Pujol, Autonomous University of Barcelona, 08916 Badalona, Spain; jrfanjul.germanstrias@gencat.cat; 6Biostatistics, Department of Basic Clinical Practice, University of Barcelona, 08036 Barcelona, Spain; 7Pediatric Infectious Diseases Research Group, Institut de Recerca Sant Joan de Déu, CIBERESP, 08950 Barcelona, Spain

**Keywords:** ventilator-associated pneumonia (VAP), lung ultrasound (LUS), procalcitonin (PCT), clinical pulmonary infection score (CPIS)

## Abstract

Ventilator-associated pneumonia (VAP) is common in Pediatric Intensive Care Units. Although early detection is crucial, current diagnostic methods are not definitive. This study aimed to identify lung ultrasound (LUS) findings and procalcitonin (PCT) values in pediatric patients with VAP to create a new early diagnosis score combined with the Clinical Pulmonary Infection Score (CPIS), the CPIS-PLUS score. Prospective longitudinal and interventional study. Pediatric patients with suspected VAP were included and classified into VAP or non-VAP groups, based on Centers of Disease Control (CDC) criteria for the final diagnosis. A chest-X-ray (CXR), LUS, and blood test were performed within the first 12 h of admission. CPIS score was calculated. A total of 108 patients with VAP suspicion were included, and VAP was finally diagnosed in 51 (47%) patients. CPIS-PLUS showed high accuracy in VAP diagnosis with a sensitivity (Sn) of 80% (95% CI 65–89%) and specificity (Sp) of 73% (95% CI 54–86%). The area under the curve (AUC) resulted in 0.86 for CPIS-PLUS vs. 0.61 for CPIS. In conclusion, this pilot study showed that CPIS-PLUS could be a potential and reliable tool for VAP early diagnosis in pediatric patients. Internal and external validations are needed to confirm the potential value of this score to facilitate VAP diagnosis in pediatric patients.

## 1. Introduction

Ventilator associated pneumonia (VAP) is the second most frequent hospital-acquired infection (HAI) in the Pediatric Intensive Care Unit (PICU) [1,2,3,4,5]. Its incidence is defined between 5–14% of ventilated pediatric patients [6,7,8,9] and is associated with increased length of stay (LOS), broad-spectrum antibiotic usage, morbidity, and mortality [10,11,12,13]. VAP suspicion should be treated promptly but not massively [14,15,16]. Common entities in critically ill patients, such as pulmonary edema, pulmonary hemorrhage, and acute respiratory distress syndrome, may overlap with pneumonia features [17,18,19], leading to an antibiotic overuse. Due to VAP incidence and the lack of an early suspicion diagnosis gold standard [20,21] there is a need for new diagnostic approaches for development.

The chest X-ray (CXR) is the most widely used image diagnosis test. The changes in radiological findings are intrinsic to most diagnostic VAP algorithms [22]. However, evidence suggests that the inherent subjectivity and shortcomings of the CXR findings make it poorly specific for ICU use [23]. The computed tomography (CT) is the gold standard [24], but its drawbacks include radiation (especially in pediatrics) and patient transfer challenges. Lung ultrasound (LUS) is a radiation-free and bedside-available alternative that has shown promise in diagnosing VAP in adults [24,25]. However, LUS has limitations as it is operator dependent and does not provide a panoramic view of both lungs or quantify pneumothorax. A recent study [15] has also found that LUS can diagnose VAP in the early stage and with high sensitivity (Sn) and specificity (Sp). Nevertheless, scarce data exists regarding its role and usefulness in pediatric VAP diagnosis, with ongoing investigations [26,27].

Prompt antibiotic indication is crucial in VAP to prevent increased morbidity and mortality [28,29]. However, empirical therapy (which accounts for half of antibiotic use in PICU), and extended treatment without VAP confirmation can lead to inappropriate and excessive antibiotic use, fostering antimicrobial resistance [30,31,32]. Moreover, patients with VAP are at a high risk for antimicrobial resistance microorganisms (AMR) infections.

The use of biomarkers like procalcitonin (PCT) and C-reactive protein (CRP) might be helpful to elucidate the etiology of bacterial respiratory infections potentially requiring antibiotics. PCT usefulness has been proven for bacterial infection diagnosis [26,33,34,35,36,37], being more Sn than CRP (92 vs. 86%) for similar Sp (73 and 70%) [34]. However, despite its role in pneumonia diagnosis, PCT alone may not confirm the diagnosis of bacterial pneumonia [38].

The Clinical Pulmonary Infection Score (CPIS) [39,40] has been proposed as a diagnosis tool for VAP in pediatric patients [41,42]. It combines various clinical, analytical, imaging, and microbiological data to calculate a score that guides VAP diagnosis. Nevertheless, subsequent studies have shown that the CPIS score has a low diagnostic performance [43], leading to VAP over-diagnose and excessive prescription of antibiotics [21].

In the adult population, a new CIPIS score incorporating PCT and/or LUS findings demonstrated to improve accuracy in VAP diagnosis [15,44,45], but a reliable diagnostic tool for early VAP diagnosis in pediatrics is required.

The aim of our study was to develop a new score based on CPIS, incorporating PCT values (CPIS-PLUS) and/or LUS (CPIS-LUS) and to analyze its accuracy for pediatric VAP diagnosis.

## 2. Materials and Methods

### 2.1. Study Design and Patients

This was a prospective longitudinal and interventional study conducted in a PICU of a tertiary pediatric hospital. The study included patients under 18 years old admitted to the PICU with VAP suspicion, between 2017–2020. Patients with underlying pathologies such as cystic fibrosis or those who were immunocompromised were excluded. Violation of the study protocol and refusal of parental consent were considered withdrawal and abandonment criteria. Written parental informed consent was mandatory. The study was conducted in accordance with the Declaration of Helsinki and approved by the local Healthcare Ethics Committee and the Institutional Review Board (code: PIC-139-16).

### 2.2. Definitions

Suspected VAP: The presence of a new respiratory deterioration after 48 h of mechanical ventilation associated with clinical signs and symptoms and laboratory parameters (temperature of <36 °C or ≥38 °C, leukocyte count of ≤4.000 or ≥11.000/mm^3^, new onset of purulent sputum or change in character sputum, tachypnea, shortness of breath) and/or abnormal respiratory auscultation sounds (hypoventilation, tubular breath sounds, murmur) [4,46].

Confirmed VAP: final diagnosis of VAP was completed based on the CDC’s definition (Appendix A) [4,46].

### 2.3. Variables

Demographic data: age, sex, medical history, previous comorbidities, and the Pediatric Risk Score of Mortality III scale (PRISM III). Microbiological data: viral determination in respiratory samples; bacterial cultures and the isolated microorganism—in bronchoaspirate (BA) or bronchoalveolar lavage (BAL) and blood culture. Evolution data: need and duration of respiratory support like noninvasive ventilation (NIV, meaning CPAP and BiPAP), conventional mechanical ventilation (CMV), and nitric oxide (NO); need and duration of hemodynamic support and extracorporeal membrane oxygenation (ECMO); length of hospital and in PICU stay (LOS). Mortality was considered any death occurring at 28 days of discharge.

### 2.4. Image Diagnostic Test

CXR was interpreted by a senior consultant radiologist who specializes in pediatrics and has dedicated his entire 15-year career to this field, with a special focus on pediatric pulmonary pathologies. The diagnosis of bacterial pneumonia (BP) was based on previous publications [47].

LUS was performed by intensive care physicians who had received special training in LUS and had at least 3 years of experience, and more than 250 LUS performed. Subjects were examined in the supine and lateral decubitus positions. A 12-Mhz lineal probe was used to systemically scan 6 areas for each hemithorax (superior and inferior of each anterior, lateral, and posterior zones), in accordance with international recommendations [48].

The LUS features considered were [11,15,49]:-The presence of B-lines and their characteristics (long, spared or confluent), and their position (peri-lesional, unilateral/bilateral).-The main lesions (consolidation): size (divided into <15 mm, 15–20 mm, >20 mm), if they were single or multiple, location (unilateral or bilateral) [49]-The presence and number (<2 or ≥2) of small subpleural consolidations (<10 mm in diameter) and their position (unilateral/bilateral) [11,15,49].-Presence and type of bronchogram and its dynamicity during breath (fix or dynamic).-Presence of pleural effusion.-Presence of pneumothorax (lung point, absence of lung sliding (M-mode).

The different patterns of aeration and diagnosis of pneumonia were based on previous publications [11,26,50,51]. BP pattern was based on the presence of lung consolidation with air bronchogram and/or the presence of whiteout lung [11,52]. Atelectasis was considered as consolidation with a tissue-like pattern with static air bronchogram [11,52,53].

### 2.5. PCT Values

PCT values were determined via LumiTest^®^PCT immune-luminometric assay (ATOM S.A.; Brahms Diagnostica GmbH, Hennigsdorf, Germany). PCT values were established based on the findings of previous similar studies conducted in adults reported in the literature [44]. The cut-off points of PCT score were categorized as <0.5 ng/mL, ≥0.5–1 ng/mL, and ≥1 ng/mL, and were assigned values of 1, 2 and 3 points, respectively. The development of these cut-off points was based on a previous clinical trial that aimed to decrease the use of antibiotic therapies in the ICU (PRORATA trial) [54].

### 2.6. Microbiological Study

Cultures were indicated according to medical criteria when bacterial infection was suspected. To differentiate between colonization and infection, the quantification of colony forming units (CFU) per milliliter was used. The BA/BAL cultures were considered positive if there was growth of >10^5^ CFU/mL for BA, and >10^4^ CFU/mL for BAL [55,56,57].

Appendix A shows threshold values for cultured specimens used in the diagnosis of pneumonia.

### 2.7. Data Management

Demographical and clinical data, and analytical and microbiological results were collected. When VAP was suspected (after a minimum of 48 h under CMV), patients underwent a chest X-ray (CXR), a lung ultrasound (LUS), and a blood test including procalcitonin (PCT) determination within the first 12 h of VAP suspicion. The CPIS score, shown in Table 1, was calculated in all patients at the same time [39]. CXR and LUS images were recorded and stored for further checks if needed.

Patients were classified into two groups; confirmed VAP and no-VAP. Different scores based on previous publications [11,15,45] were defined and calculated in both groups.

The accuracy of each test (CPIS, CPIS-LUS and CPIS-PLUS) was calculated to identify the CPIS score with better accuracy. The values of the LUS test were assigned in accordance with previous reports [58].

### 2.8. Outcomes

The outcome was to assess the accuracy of the different scores: CPIS, CPIS-LUS, and CPIS-PLUS for VAP early diagnosis.

### 2.9. Statistical Analysis

Descriptive statistical analysis of data was performed using the compareGroups R package [59]. Quantitative variables were described by mean and standard deviation or median and interquartile range (IQR), depending on the variable distribution. Frequencies and percentages were used for qualitative variables. The proportions of categorical variables were compared using the X2 test or Fisher’s exact test, and odds ratios were estimated to assess the magnitude of the association. Comparison of means of continuous variables was assessed by the *t*-test or Mann–Whitney U test.

The score assignment to construct the CPIS-PLUS tests was based on the log-odds ratios criteria, as suggested in [59]. Diagnostic accuracy and predictive ability of the CPIS-PLUS tests were assessed by estimation of the sensitivity (Sn), specificity (Sp), positive and negative predictive values (PPV and NPV), and area under the ROC curve (AUC) using the ThresholdROC v.2.9 [60,61] and pROC R packages v.1.18 [61]. Optimum cut-off points were estimated by maximizing the Youden index, whilst paired AUC were compared using DeLong’s approach. A *p*-value of <0.05 was considered statistically significant.

## 3. Results

A total of 108 patients had eligibility criteria for suspected VAP and were enrolled. There were 56 (51.9%) males, and the median age was 8.4 months old (IQR 0.76–57). The median PICU LOS was 23 days (IQR 16–60.5), and the median hospitalization LOS resulted in 43 days (IQR 25–108). Of the 108 patients, 51 (47.2%) patients were finally diagnosed with VAP, which represents a mean rate of 4.8/1000 mechanical ventilation days along the study period. The CPIS score resulted in a mean of 6.63 (standard deviation (SD) 1.74) vs. 5.09 (SD 2.12) points (*p* < 0.01), for patients with and without VAP, respectively. Demographic and clinical characteristics were similar in both groups (VAP and no-VAP); Table 2.

LUS was performed on 84 patients. Appendix A compares CXR and LUS findings in a patient with confirmed VAP.

Regarding LUS findings, different features were found in patients with VAP vs. no-VAP final diagnosis:
(1)Lobar consolidations were found in 47 (69.1%) vs. 21 (30.9%) patients; and subpleural consolidations in one (12.5%) vs. in seven (87.5%) cases, *p* < 0.001.(2)Unilateral consolidations were found in 41 (73.2%) vs. 15 (26.8%) patients; and bilateral consolidations in seven (35%) vs. thirteen (65%) cases, *p* < 0.001.(3)Consolidations > 20 mm were found in 39 (76.5%) vs. 12 (23.5%) patients, and those between 15–20 mm in eight (47.1%) vs. nine (52.9%) cases, *p* < 0.001.(4)Regarding the presence and type of bronchogram: dynamic bronchogram (DB) was seen in thirty-eight (95%) vs. two (5%) patients; and static bronchogram (SB) in 10 (34.5%) vs. 19 (65.5%) cases (*p* < 0.001).(5)PCT resulted with a median of 0.47 (IQR 0.17–1.32) in VAP patients vs. 0.21 (IQR 0.10–0.79) in those without VAP, *p* = 0.300.

LUS findings and PCT values with its log-odds ratios are described in Table 3.

The CPIS-PLUS score was designed considering the OR of each LUS finding and PCT values in both VAP and non-VAP patients, assigning different punctuation according to these results. The sub-analysis of the different LUS features demonstrated that the best LUS feature to predict VAP was the presence of dynamic air bronchogram. Table 4 shows the new proposed Score.

From the total sample (n = 108), a CPIS cut-off of 6 points showed an AUC of 0.70 (95% CI: 0.61–0.80), Sn of 78% (95% CI: 64–88%), Sp in 56% (95% CI: 42–69%), PPV in 62% (95% CI: 49–73%), and a negative NPV of 74% (95% CI: 59–85%) for VAP diagnosis.

When analyzing patients with LUS (n = 84), a CPIS cut-off of 6 points showed an AUC of 0.61 (95% CI: 0.50–0.75) with a Sn of 78 (95% CI: 63–88%) and Sp of 46% (95% CI: 29–63%). When combining CPIS with LUS findings, CPIS-LUS, the AUC increased to 0.86, *p* < 0.01. Then, when adding PCT, CPIS-PLUS, Sn increased to 80%, Sp decreased to 73%, and the AUC resulted the same (0.86, *p* = 0.79).

Analyzing LUS findings alone, using LUS ≥ 7 and CPIS-LUS ≥ 12 as thresholds for positive results, the score showed better AUC (0.91 vs. 0.86, *p* 0.27) and better Sn (81% vs. 76%) for LUS alone.

The addition of the PCT values to CPIS did not improve the diagnostic accuracy.

Accuracy results for LUS scores are summarized in Table 5.

Comparison of AUC of different scores are showed in Figure 1.

## 4. Discussion

This study highlights LUS accuracy for VAP early diagnosis, with higher results in AUC than previously validated scores in pediatric populations such as CPIS. Even when adding blood biomarkers to those scores, LUS accuracy remains superior.

The combination of clinical, microbiological, and radiological findings has shown to be a valuable diagnostic tool for VAP, presenting higher accuracy than when these factors are used separately [62]. The CPIS score, widely implemented for VAP diagnosis, incorporates CXR findings for radiological assessment, which can sometimes lead to VAP overdiagnosis. In the adult population, several scores have been published in the literature, such as the Chest Echography and Procalcitonin Pulmonary Infection Score (CEPPIS) [44], Ventilator-associated Pneumonia Lung Ultrasound (VPLUS) [15], and Sono-Pulmonary Infection Score (SPIS) [45]. Overall, studies aimed to improve VAP early diagnosis and appropriate empirical antibiotic treatment. Moreover, all substituted CXR for LUS, making LUS the bedside imaging test, and LUS findings the radiological features integrated into VAP scores. The first two scores also included direct gram examination of TA, allowing an earlier diagnosis of VAP and showing superior diagnostic accuracy compared to CPIS alone. But in our study of the pediatric population LUS alone was the most useful.

In the CEPPIS study [44], Zagli et al., showed that a Clinical-LUS score based on the presence of lobar consolidation or small anterolateral subpleural consolidations performed better than CPIS alone. Their results showed that a CEEPIS > 5 was significantly better for VAP prediction than CPIS, resulting in an AUC of 0.829 vs. 0.616, respectively. Like this study, the CPIS-PLUS score includes clinical, microbiological, PCT, and LUS features. In line with their results, CPIS-PLUS ≥ 12 had high diagnostic accuracy with better Sn, Sp, and an AUC of 0.86 vs. AUC of 0.61 for CPIS. In the VPLUS study [15], Mongodi et al. found that the combination of dynamic linear/arborescent bronchogram, or the presence of equal or more than two areas with subpleural consolidations, and a positive gram stain examination, showed an AUC of around 0.8. However, they only considered purulent secretion as the only clinical criteria, a subjective item. The results showed that AUC for VPLUS-EA gram and CPIS were, respectively, 0.832 (0.737–0.903) vs. 0.576 (0.473–0.675). Finally, in the SPIS study, Samanta et al. [45] agreed that the best LUS finding to diagnose VAP was the presence of dynamic air bronchogram, followed by equal or more than two areas with subpleural consolidation, or lobar consolidation, or the presence of equal or more than one subpleural, and equal or more than one lobar consolidation. SPIS showed higher Sn (87.5%), Sp (76.7%), and PPV (90.9%) with better AUC (0.913) compared to CPIS alone, including microbiological tests in both cases. These findings were homogeneous with the results of our study, where the sub-analysis of the different LUS features demonstrated that the best LUS feature to predict VAP was a dynamic air bronchogram. The AUC for the CPIS-LUS score was 0.86 vs. 0.61 for CPIS.

In our sample, when considering LUS findings alone compared to CPIS-LUS, the highest score resulted for the LUS, which showed the highest Sn, similar Sp, and higher AUC. The CPIS score had good Sn, but not Sp. However, when LUS findings were added to these results, the Sp rate significantly increases.

LUS can improve early diagnosis of lung diseases in critically ill children, being more sensitive than CXR and comparable to CT, without radiation exposure. Although it is operator dependent, different studies have shown that even beginners can perform it with high precision, showing adequate interobserver agreement in image interpretation [63,64]. Obesity and subcutaneous emphysema are the primary patient-related factors that can limit the effectiveness of LUS. However, pediatric patients are usually good candidates for LUS as they have less subcutaneous tissue. There is a limitation to using LUS for the location of pulmonary pathology since it cannot visualize the etiologies that do not extend to the pleural surface. However, more than 95% of pulmonary pathological processes have a pleural component in both adults and pediatric patients, making ultrasound useful in most cases [63,65].

The efficacy of LUS in diagnosing pneumonia in children has been demonstrated in previous studies. Buonseno et al. [11] described that multiple and bilateral consolidations smaller than 15 mm were more commonly observed in viral pneumonia (VP), in contrast to those over 40 mm that suggested bacterial infection with statistically significant differences (*p* < 0.001). Likewise, Berce et al. [49] reported similar results, and they concluded that consolidations in VP were significantly smaller, with a median diameter of 15 mm, compared to 20 mm in atypical BP and equal or over 30 mm in BP. These findings were comparable with the results of our study, where the presence of lobar consolidations over 20 mm (76.5%, OR 22.8), unilateral consolidation (73.2%, OR 19.1), and the presence of dynamic bronchogram (95%, OR 266) were significantly associated with BP. Regarding LUS findings in VAP, Mongodi et al. [15] showed that a dynamic air bronchogram was one of the best signs to predict VAP in adulthood. On the other hand, a study of neonates [66] showed that the presence of lung consolidation with air bronchogram had the highest Sn, Sp, and accuracy for VAP diagnosis. In line with these findings, our study also found that the presence of dynamic air-bronchogram was more frequent in patients with a VAP diagnosis.

Regarding the addition of biomarker values, PCT has been shown to be a valuable biomarker for the diagnosis and follow-up of community-acquired pneumonia. A recent pediatric population study [26] demonstrated that a non-typical bacterial consolidation pattern detected on LUS with the addition of PCT improved the diagnosis accuracy. However, PCT alone might not be a reliable diagnostic biomarker for VAP, consistent with our results. In the adult population, the CEPPIS study [44] proposed to replace LUS for CXR and PCT values for leukocytes. The authors demonstrated that CEPPIS had higher Sn than CPIS alone, resulting in 80.5% vs. 40%, respectively—and similar Sp of 85.2% vs. 83.3%, respectively, with higher AUC (0.83 vs. 0.62), *p* < 0.001. Afterward, these findings were confirmed by other authors. Zhou J. et al. [67] showed better Sp (85.5%) when combining a positive LUS examination with a PCT value ≥ 0.25 ng/mL, significantly higher than CPIS alone (Sp 57.9%), *p* < 0.01.

Other authors found no differences in the score accuracy results when adding PCT values. Gibot et al. [68] found no difference in serum PCT levels between VAP and non-VAP cases. Also, Mongody et al. [15] omitted PCT values in the PLUS score, as they suggested that lower levels were associated with VAP. They found a statistically significant difference in PCT levels between VAP and non-VAP cases (1.0 vs. 5.0 ng/mL; *p* < 0,05), indicating a potentially low impact of PCT on early diagnosis. Similar results have been reported in other prospective studies [45,69]. In line with these results, our study also observed that PCT value alone did not effectively discriminate between VAP and non-VAP cases, although there was a trend for higher PCT levels in patients with VAP vs. non-VAP (0.47ng/mL vs. 0.21ng/mL, *p* = 0.300). Moreover, when adding the PCT value to the score, CPIS-PLUS, the Sn (80% vs. 76%), Sp (73% vs. 77%), and AUC (0.86 vs. 0.86) were similar than when combining CPIS and LUS findings, and lower when compared to LUS findings alone; Sn (80% vs. 81%), Sp (73% vs. 74%), with AUC (0.86 vs. 0.91). This result could be since the integration of LUS allowed for earlier detection of VAP, potentially limiting the timeframe required for PCT rising as a blood biomarker. Patients in the PICU are closely monitored, and therefore, clinical suspicion of VAP is identified early, and LUS is performed immediately. As a result, LUS seems to have contributed more than PCT to the early diagnosis of VAP. In summary, while PCT has proven to be a valuable biomarker for CAP diagnosis and follow-up, its utility as a diagnostic biomarker for VAP is limited. Studies have shown mixed results regarding the effectiveness of PCT alone in discriminating between VAP and non-VAP cases. Integrating PCT with other clinical and microbiological factors, such as CPIS and LUS, may improve the accuracy of VAP diagnosis compared to CPIS alone.

Improving the VAP early diagnostic is relevant because it is the second most prevalent HAI among the pediatric population. The results of our study showed a VAP frequency of 47% from those patients who had VAP suspicion, which aligns with a similar study conducted in the adult’s population where a prevalence of 64% of VAP was reported [15]. In ventilated PICU patients, VAP had a rate that ranged from 2.9 to 21.6 per 1000 ventilator days in different studies [70,71,72]. In our sample, the rate was in line with these previous publications. Interestingly, another study demonstrated that applying different diagnostic criteria resulted in VAP incidence rates varying from 4 to 42% within the same patient population [73].

The present study has several limitations. Firstly, it was conducted in a single center, in which pediatricians are skilled well in LUS performance, which could potentially affect the reproducibility of these high accuracy results. Secondly, since there is no good gold standard diagnostic test for VAP, it is difficult to establish comparisons. However, this study aimed to assess the improvement of early VAP diagnosis based compared to CPIS score, a standardized and validated score. As previously mentioned, the final diagnosis of VAP was based on CDC criteria.

Further multicentric studies will be needed to validate the new CPIS-PLUS score.

## 5. Conclusions

The new CPIS-PLUS score, combining PCT and LUS findings with CPIS, may provide a practical clinical approach for early VAP diagnosis and could lead to commence empirical antibiotics in patients with VAP suspicion.

LUS findings resulted with better diagnostic accuracy, especially increasing the specificity for VAP diagnosis. LUS findings resulted even better than the CPIS-PLUS score, at least in this study where PICU pediatricians are highly trained and experienced in LUS.

The results of this study, lead and force the authors to design and conduct an external validation of the CPIS-PLUS score throughout a multicenter study.

## Figures and Tables

**Figure 1 children-11-00592-f001:**
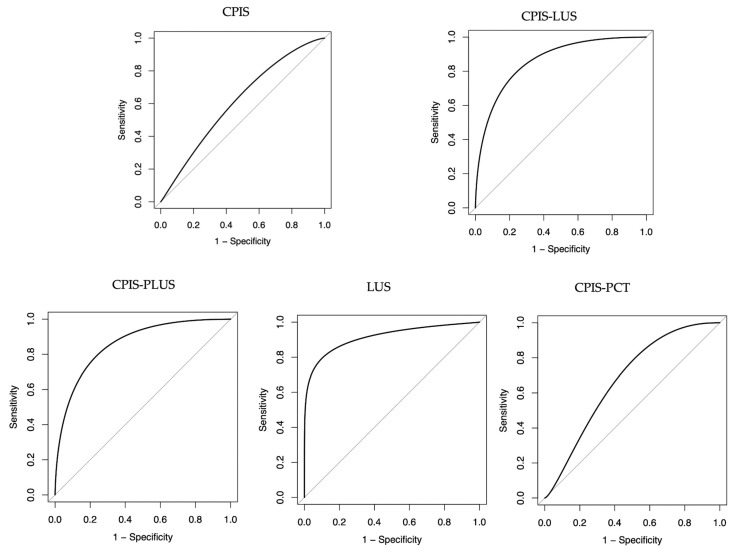
AUC CPIS, CPIS-LUS, CPIS-PLUS, LUS, and CPIS-PCT.

**Table 1 children-11-00592-t001:** CPIS [39] Clinical Pulmonary Infection Score.

Parameter	Points		
	0	1	2
Temperature, °C	≥36.5 and ≤38.4	≥38.5 and ≤38.9	≤36 or ≥39
Blood leukocytes, WBC/mm^3^	≥4.000 and ≤11.000	<4.000 or >11.000	<4.000 or >11.000 and bands forms ≥500
Oxygenation: PaO_2_/FiO_2_	>240 or presence of ARDS	----	≤240 and absence of ARDS
Tracheal secretion	Absent or minimal (<14 aspirations/day)	Non purulent (≥14 aspirations/day)	Purulent
Pulmonary radiography	No infiltrate	Patchy o diffuse infiltrate	Localized infiltrate
Culture/microscopy of tracheal aspirate	Negative culture	Positive culture	Positive culture and microscopy ^1^ (same pathogenic bacteria seen on Gram stain)

Abbreviations: ARDS, acute respiratory distress syndrome, is defines as a PaO_2_/FiO_2_ ≤ 200, pulmonary artery wedge pressure < 18 mm Hg and acute bilateral infiltrates; PaO_2_/FiO_2_, ratio of partial pressure of arterial oxygen to the fraction of inspired oxygen. ^1^ Positive microscopy result means positive result on direct Gram stain examination of endotracheal aspirate. Positive culture means positive result on the culture of endotracheal aspirate with colony-forming unit ≥ 10^5^ or BAL ≥ 10^4^.

**Table 2 children-11-00592-t002:** Demographic and clinical variables.

	Total n = 108	VAP Group n = 51	No-VAP Group n = 57	*p* Value
Gender (male), n (%)	56 (51.9)	23 (45)	33 (57.8)	0.184
Age (months) median (IQR)	8.4 (0.76–57)	15 (1.58–63.25)	6.31 (3.09–54.7)	0.827
Weight (kg), median (IQR)	8 (4.6–19.25)	9 (4.5–20)	7.2 (4.65–17)	0.622
Comorbidities, n (%)				0.365
None	37 (34.3)	20 (39)	17 (29.8)	
Respiratory	26 (24.1)	10 (19.6)	16 (28)	
Cardiovascular	21 (19.4)	9 (17.6)	12 (21)	
Infectious diseases	1 (0.9)	1 (1.96)	0 (0)	
Neurological	10 (9.3)	3 (5.88)	7 (12.3)	
Hematology-oncology	5 (4.6)	2 (3.92)	3 (5.26)	
Other	8 (7.4)	6 (11.8)	2 (3.5)	
Reason for admission, n (%)				0.065
Infection ^1^	32 (26.9)	21 (41.2)	11 (19.3)	
Trauma	8 (7.4)	5 (9.8)	3 (5.26)	
CV surgery/heart failure	18 (16.7)	7 (13.7)	11 (19.3)	
Oncology	4 (3.7)	0 (0)	4 (7)	
Surgery (Abd, trauma, NS)	21 (19.4)	8 (15.7)	13 (22.8)	
Other	25 (23.1)	10 (19.6)	15 (26.3)	
Severity upon admission,				
PRISM III, median (IQR)	3 (0–9.75)	4 (0–8)	3 (0–10)	0.249
Length of stay				
PICU (days), median (IQR)	23 (16–60.5)	23 (16–35)	27 (12.5–84)	0.571
Hospitalization (days), median (IQR)	43 (25–108)	35 (24–66)	69 (25.5–127)	0.128
Clinical symptoms/signs n (%)				
Fever	74 (68.5)	38 (74.5)	36 (63.2)	0.205
Tracheal Secretions Absent Non-purulent Purulent	44 (40.7) 47 (43.5) 17 (15.7)	12 (23.5) 24 (47.1) 15 (29.4)	32 (56.1) 23 (40.4) 2 (3.5)	0.001
Blood test				
Leukocytes, median (IQR)	12.400 (8.500–16.200)	12.400 (8.500–16.200)	12.600 (8.500–16.200)	0.568
CRP mg/L, median (IQR)	58.8 (20.85–98.5)	56.35 (24–105)	60 (12–93)	0.666
PCT ng/mL, median (IQR)	0.36 (0.16–1.42)	0.44 (0.17–1.31)	0.25 (0.11–1.77)	0.549
Antibiotic therapy, n (%)	89 (82.4)	51 (100)	38 (66.7)	*p* < 0.01
Days of antibiotic, median (IQR)	7 (5–7)	7 (7–9)	5 (0–7)	*p* < 0.01
CMV days, median (IQR)	9 (3–15)	12 (8–16)	6 (0–11)	*p* < 0.01
Inotropic support, n (%)	37 (34.3)	21 (41.2)	16 (28)	0.152
Exitus, n (%)	5 (4.6)	2 (3.9)	3 (5.3)	1
CPIS (mean ± SD)	5.81 ± 2.09	6.63 ± 1.74	5.09 ± 2.12	<0.001

Abbreviations: Abd, abdominal surgery; CMV, conventional mechanical ventilation; CRP, C-reactive protein; CV, cardiovascular; IQR, interquartile range; NS, neurosurgery; PICU, pediatric intensive care unit; PCT, procalcitonin. ^1^ Infection: meningitis, pneumonia, sepsis, bronchiolitis.

**Table 3 children-11-00592-t003:** LUS findings, PCT values, and its log-odds ratio in all patients, and in both VAP and No-VAP groups.

LUS Findings	Total n = 84	VAP n = 49	No-VAP n = 35	OR [IQR]	*p*-Value OR	*p*-Value Overall
Type of consolidation n (%)						<0.001
No consolidation	8 (9.52%)	1 (12.5%)	7 (87.5%)			
Supleural	8 (9.52%)	1 (12.5%)	7 (87.5%)	1.00 [0.05–19.4]	1	
Lobar	68 (81%)	47 (69.1%)	21 (30.9%)	15.7 [1.81–136]	0.003	
Laterality n (%)						<0.001
No consolidation	8 (9.52%)	1 (12.5%)	7 (87.5%)			
Bilateral	20 (23.8%)	7 (35%)	13 (65%)	3.77 [0.38–37.1]	0.281	
Unilateral	56 (66.7%)	41 (73.2%)	15 (26.8%)	19.1 [2.17–169]	0.002	
Size n (%)						<0.001
No consolidation	8 (9.52%)	1 (12.5%)	7 (87.5%)			
<15 mm	8 (9.52%)	1 (12.5%)	7 (87.5%)	1.00 [0.05–19.4]	1	
≥15 mm–20 mm	17 (20.2%)	8 (47.1%)	9 (52.9%)	6.22 [0.62–62.2]	0.119	
>20 mm	51 (60.7%)	39 (76.5%)	12 (23.5%)	22.8 [2.54–204]	0.001	
Bronchogram n (%)						<0.001
No bronchogram	15 (17.9%)	1 (6.67%)	14 (93.3%)			
Static bronchogram	29 (34.5%)	10 (34.5%)	19 (65.5%)	7.37 [0.84–64.4]	0.048	
Dynamic bronchogram	40 (47.6%)	38 (95%)	2 (5%)	266 [22.3–3168]	<0.001	
PCT ng/mL (median [IQR])	0.37 [0.16–1.31]	0.47 [0.17–1.32]	0.21 [0.10–0.79]	0.95 [1.01–0.89]	0.125	0.3

Abbreviations: IQR, interquartile range; PCT, procalcitonin; OR, Odds Ratio.

**Table 4 children-11-00592-t004:** CPIS-PLUS Score.

CPIS-PLUS	0	1	2	3
Temperature, °C	≥36.5 and <38.4	≥38.5 and <38.9	<36 or ≥39	-
Leukocytes/mm^3^	≥4.000 and ≤11.000	<4.000 or >11.000	<4.000 or >11.000 and bands forms ≥500	-
PaO_2_/FiO_2_	>240 or ARDS	----	≤240 and no evidence of ARDS	-
Tracheal secretion	Absent or minimal (<14 aspirations/day)	Non purulent (≥14 aspirations/day)	Purulent	-
Culture or gram of tracheal aspirate	Negative	Positive	Positive culture and microscopy (same pathogenic bacteria seen on Gram stain)	-
Procalcitonin, ng/mL	<0.5	≥0.5 and <1	≥1	-
LUS findings				
Type of consolidation	No consolidation	Subpleural	Lobar	-
Size of consolidation	<15 mm	15–20 mm	>20 mm	-
Laterality	-	Bilateral	Unilateral	-
Bronchogram	-	SB	-	DB

Abbreviations: ARDS, acute respiratory distress syndrome, is defines as a PaO_2_/FiO_2_ ≤ 200, pulmonary artery wedge pressure < 18 mm Hg and acute bilateral infiltrates; DB, dynamic bronchogram, LUS: Lung ultrasound. PaO_2_/FiO_2_: ratio of partial pressure of arterial oxygen to the fraction of inspired oxygen; SB, static bronchogram. Positive microscopy result means positive result on direct Gram stain examination of endotracheal aspirate. Positive culture means positive result on the culture of endotracheal aspirate with colony-forming unit ≥ 10^5^ or BAL ≥ 10^4^.

**Table 5 children-11-00592-t005:** Diagnostic accuracy of the different test combinations (n = 84).

	AUC	Sensibility (%)	Specificity (%)	PPV (%)	NPV (%)
CPIS > 6	0.61 (0.50–0.75)	78 (63–88)	46 (29–63)	67 (53–78)	59 (39–77)
CPIS-LUS ≥ 12	0.86 (0.76–0.92)	76 (61–86)	77 (59–89)	82 (67–91)	69 (52–82)
CPIS-PLUS ≥ 12	0.86 (0.76–0.92)	80 (65–89)	73 (54–86)	81 (67–91)	71 (52–84)
LUS ≥ 7	0.91 (0.78–0.95)	81 (67–91)	74 (56–87)	82 (67–91)	74 (56–87)
CPIS-PCT ≥ 7	0.67 (0.57–0.78)	60 (45–73)	60 (45–73)	60 (45–73)	60 (45–73)

Estimate (95% CI).

## Data Availability

The raw data supporting the conclusions of this article will be made available by the authors on request. The data are not publicly available due to privacy.

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
