# Peer review of "The Clinical Pulmonary Infection Score Combined with Procalcitonin and Lung Ultrasound (CPIS-PLUS), a Good Tool for Ventilator Associated Pneumonia Early Diagnosis in Pediatrics"

_children, 2024, doi:10.3390/children11050592_

Round 1
Reviewer 1 Report
Comments and Suggestions for Authors
The study is devoted to a very urgent problem - improving the diagnosis of ventilator-associated pneumonia. It is known that there is still no gold standard either for adults or children, which leads to hypo- and hyperdiagnosis and, accordingly, unjustified prescription of antibiotics.
The CPIS has been studied in a number of studies in the paediatric population, demonstrating acceptable diagnostic accuracy, but the task of improving both its sensitivity and specificity remains unresolved. This, in fact, determines the relevance of the research.
At the same time, I have questions about both the design of the study and the results, which I would like to see clarified:
1. Materials and Methods
- Why was the definition of СDC chosen to verify the diagnosis of VAP and what specific criteria were used? This, in my opinion, should be added to the supplementary materials
- For what purpose was urine culture used?
- It would be useful to add how the lung ultrasound was still interpreted and what was put into the model - an overall summary score or specific patterns/patterns of change?
- You declare that the following interpretation of PCT was used "A PCT value of <1 ng/ml was considered normal. A PCT value ≥1 ng/ml was considered to indicate bacterial infection"
However, quantitative values and other cut-off points appear in the results. This needs to be clarified.
- Table 1 - there is an error in the temperature level giving 3 points
2. Results
- The sub-analysis of the different LUS features demonstrated that the best LUS feature to predict VAP was the presence of dynamic air bronchogram - could you please clarify what is the point of including other findings on lung ultrasound into the model if one is most significant?
3. Conclusion
According to the results of your study, it seems that lung ultrasound was the best tool for diagnosing VAP. Then is it still reasonable to recommend the use of CPIS-PLUS score? Can you provide additional reasoning?
In addition, I would like to draw the attention of the authors to the fact that not all references are consistent with the claims made, and not all claims referring to evidence are supported by references. For example, in the introduction:
The clinical pulmonary infection score (CPIS) has been proposed as a diagnosis tool for VAP in pediatric patients[39,40].
- There are studies examining the diagnostic accuracy of CPIS in the paediatric population that you do not cite.
Nevertheless, subsequent studies have shown that the CPIS score has a low diagnostic performance, leading to VAP over-diagnose and excessive prescription of antibiotics.
- No reference to specific studies
English is generally good, terminology is consistent with common terminology, needs minor stylistic edits
Author Response
RESPONSE TO THE REVIEWER COMMENTS
Dear editor and reviewers, thank you very much for giving us the opportunity to improve the manuscript. We hope we have clearly answered all the questions. Please, let us know if further changes are required. All changes in the manuscript are highlighted in yellow.
Reviewers' Comments to Authors
REVIEWER 1
Comments to the Author
The study is devoted to a very urgent problem - improving the diagnosis of ventilator-associated pneumonia. It is known that there is still no gold standard either for adults or children, which leads to hypo- and hyperdiagnosis and, accordingly, unjustified prescription of antibiotics.
The CPIS has been studied in a number of studies in the paediatric population, demonstrating acceptable diagnostic accuracy, but the task of improving both its sensitivity and specificity remains unresolved. This, in fact, determines the relevance of the research.
MAJOR COMENTS
At the same time, I have questions about both the design of the study and the results, which I would like to see clarified:
- Materials and Methods
- Why was the definition of СDC chosen to verify the diagnosis of VAP and what specific criteria were used? This, in my opinion, should be added to the supplementary materials
Dear reviewer, thank you very much for your suggestion. In our Pediatric Intensive Care Unit (PICU), we use the Centers for Disease Control and Prevention (CDC) criteria, which are internationally recognized and regularly reviewed and updated based on the latest scientific evidence.
In 2013, new CDC definitions were published, which changed the concept from VAP to ventilator-associated events (VAEs). It was emphasized that respiratory worsening was a guide sign to identify and monitor all the processes that cause respiratory deterioration (with or without an infection cause) in the ventilated patient. If the respiratory sample obtained from the patient during that respiratory worsening met the proposed microbiological criteria, it was defined as possible ventilator-associated pneumonia (PVAP).
We believe that following established guidelines is crucial to ensure the best possible care for our patients. To help better understand the definitions, we have added two tables in the Supplementary Material that focus on the specific diagnosis criteria of VAP. Page 12, line 404
Table S1. CDC diagnostic criteria
Imaging Test Evidence |
Signs/Symptoms |
Two or more serial chest imaging test results with at least one of the following:
New and persistent or progressive and persistent - Infiltrate - Consolidation - Cavitation - Pneumatoceles, in infants ≤1 year old
Note: In patients without underlying pulmonary or cardiac disease (such as respiratory distress syndrome, bronchopulmonary dysplasia, pulmonary edema, or chronic obstructive pulmonary disease), one definitive imaging test result is acceptable.
|
For ANY PATIENT, at least one of the following: - Fever (> 38.0°C or > 100.4°F) - Leukopenia (≤ 4000 WBC/mm3) or leukocytosis (≥ 12,000 WBC/mm3) - For adults ≥ 70 years old, altered mental status with no other recognized cause And at least two of the following (from separate bullets): - New onset of purulent sputum (*) or change in character of sputum (color, consistency, odor, and quantity), or increased respiratory secretions, or increased suctioning requirements - Dyspnea, or tachypnea (**), or new onset or worsening cough - Rales (crackles) or bronchial breath sounds - Worsening gas exchange (for example, O2 desaturations [for example, PaO2/FiO2 ≤ 240], increased oxygen requirements, or increased ventilator demand) |
ALTERNATE CRITERIA, for infants ≤ 1 year old: Worsening gas exchange (for example, O2 desaturations [for example, pulse oximetry < 94%], increased oxygen requirements, or increased ventilator demand) And at least three of the following (from separate bullets): - Temperature instability - Leukopenia (≤ 4000 WBC/mm3) or leukocytosis (≥ 15,000 WBC/mm3) and left shift (≥ 10% band forms) - New onset of purulent sputum (*) or change in character of sputum (color, consistency, odor, and quantity), or increased respiratory secretions, or increased suctioning requirements - Apnea, tachypnea (**), nasal flaring with retraction of chest wall, or nasal flaring with grunting - Wheezing, rales (crackles), or rhonchi - Cough - Bradycardia (< 100 beats/min) or tachycardia (> 170 beats/min)
|
|
ALTERNATE CRITERIA, for child > 1 year old or ≤ 12 years old, at least three of the following (from separate bullets): - Fever (> 38. 0°C or > 100. 4°F) or hypothermia (< 36. 0°C or < 96.8°F) - Leukopenia (≤ 4000 WBC/mm3) or leukocytosis (≥ 15,000 WBC/mm3) - New onset of purulent sputum (*) or change in character of sputum (color, consistency, odor, and quantity), or increased respiratory secretions, or increased suctioning requirements - Dyspnea, or apnea, or tachypnea (**), or new onset or worsening cough - Rales (crackles) or bronchial breath sounds - Worsening gas exchange (for example, O2 desaturations [for example, pulse oximetry < 94%], increased oxygen requirements, or increased ventilator demand) |
* Purulent sputum is defined as secretions from the lungs, bronchi, or trachea that contain ≥ 25 neutrophils and ≤ 10 squamous epithelial cells per low power field (x100).
** In adults, tachypnea is defined as respiration rate > 25 breaths per minute. Tachypnea is defined as > 75 breaths per minute in premature infants born at < 37 weeks gestation and until the 40th week; > 60 breaths per minute in patients < 2 months old; > 50 breaths per minute in patients 2- 12 months old; and > 30 breaths per minute in children > 1 year old.
Table S2. Threshold values for cultured specimens used in the diagnosis of pneumonia
Specimen collection/techinque |
Values |
Lung tissue† |
≥ 104 CFU/g tissue |
Bronchoscopically (B) obtained specimens |
|
Bronchoalveolar lavage (B-BAL) |
≥ 104 CFU/ml |
Protected BAL (B-PBAL) |
≥ 104 CFU/ml |
Protected specimen brushing (B-PSB) |
≥ 103 CFU/ml |
Nonbronchoscopically (NB) obtained (blind) specimens |
|
NB-BAL |
≥ 104 CFU/ml |
NB-PSB |
≥ 103 CFU/ml |
Endotracheal aspirate (ETA) |
≥ 105 CFU/ml |
CFU = colony forming units, g = gram, ml = milliliter
†Lung tissue specimens obtained by either open or closed lung biopsy methods. For post-mortem specimens, only lung tissue specimens obtained by transthoracic or transbronchial biopsy that are collected immediately post-mortem are eligible for use.
- For what purpose was urine culture used?
Dear reviewer, thank you very much for your question. Critical pediatric patients often require multiple devices including endotracheal tube, central lines and urinary catheters. When these patients show signs of clinical decline or fever, healthcare professionals often collect samples, such as blood culture, urine culture, and respiratory secretion culture, to screen for infections. However, in the context of diagnosing VAP, urine culture may not be a relevant factor and can be eliminated. Page 3, line 112
- It would be useful to add how the lung ultrasound was still interpreted and what was put into the model - an overall summary score or specific patterns/patterns of change?
The LUS was interpreted following international recommendations, as mentioned in line 127, page 3.
The LUS features considered were[11,15,50]:
- The presence of B-lines and its characteristics (long, spared or confluent), and its position (peri-lesional, unilateral/bilateral).
- The main lesion (consolidation): size (divided into < 15 mm, 15-20 mm, > 20 mm), if they were single or multiple, location (unilateral or bilateral)[50]
- The presence and number (<2 or ≥ 2) of small subpleural consolidations (< 10 mm in diameter) and its position (unilateral/bilateral) [11,15,50].
- Presence and type of bronchogram and its dynamicity during breath (fix or dynamic).
- Presence of pleural effusion.
- Presence of pneumothorax (lung point, absence of lung sliding (M-mode).
The different patterns of aeration and diagnosis of pneumonia were based on previous publications[11,26,51,52]. BP pattern was based on the presence of lung consolidation with air bronchogram and/or the presence of whiteout lung[11,53]. Atelectasis was considered as consolidation with a tissue-like pattern with static air bronchogram[1153,54].
Please, let us know if you consider that further details are needed to clarify your question.
- You declare that the following interpretation of PCT was used "A PCT value of <1 ng/ml was considered normal. A PCT value ≥1 ng/ml was considered to indicate bacterial infection"
However, quantitative values and other cut-off points appear in the results. This needs to be clarified.
We appreciate the reviewer’s comment and we agree with it. We apologize if the manuscript is not clear enough. The established PCT values were based on prior research reported in the literature. In particular, Zagli et al. conducted a study aimed at developing a scoring system for the early diagnosis of ventilator-associated pneumonia (VAP) in adults, which included LUS and PCT. The cut-off points of PCT in the CEPPIS score were categorized as < 0.5 ng/ml, ³0.5-1 ng/ml, and ³1 ng/ml, and were assigned values of 1, 2 and 3 points respectively. The development of these cut-off points was based on a previous clinical trial that aimed to reduce the use of antibiotic therapies in the ICU (PRORATA trial). However, we acknowledge that the methodology was described more generally. Therefore, in order to be as accurate as possible, we have provided more detailed information, as well as updated both bibliographic citations. Page 3-4, line 145
PRORATA trial
Bouadma L, Luyt CE, Tubach F, et al. Use of procalcitonin to reduce patients’ exposure to antibiotics in intensive care units (PRORATA trial): a multicentre randomised controlled trial. The Lancet. 375:463-474. doi:10.1016/S0140
- Table 1 - there is an error in the temperature level giving 3 points
Parameter |
Points |
|
||
​ |
0​ |
1​ |
2​ |
|
Temperature, ºC​ |
≥36.5 and £ 38.4​ |
≥ 38.5 and £ 38.9​ |
£36 o ≥39​ |
|
Blood leukocytes, WBC/mm3​ |
≥ 4.000 and ≤ 11.000​ |
< 4.000 or > 11.000​ |
​ < 4.000 or > 11000 and ​ bands forms ≥ 500​ |
|
Oxygenation: PaO2/FiO2​ |
>240 or presence of ARDS​ |
----​ |
≤ 240 and absence of ARDS​ |
|
Tracheal secretion​ |
Absent​ or minimal (<14 aspirations/day) |
Non purulent (≥ 14 aspirations/day) |
Purulent |
|
Pulmonary radiography |
No infiltrate |
Patchy o diffuse infiltrate |
Localized infiltrate |
|
Culture/microscopy of tracheal aspirate​ |
Negative culture |
Positive culture |
Positive culture and microscopy1(same pathogenic bacteria seen on Gram stain) |
|
Dear reviewer, thank you very much for your suggestion. This is Table 1 as appears in the manuscript. In this document there is no parameter with 3 points. Page 4, line 171.
Table 1. CPIS [39] Clinical Pulmonary Infection Score.
- Results
- The sub-analysis of the different LUS features demonstrated that the best LUS feature to predict VAP was the presence of dynamic air bronchogram - could you please clarify what is the point of including other findings on lung ultrasound into the model if one is most significant?
Dear reviewer, thank you very much for your questions. Several studies have been conducted to identify effective ultrasound items for predicting VAP. Among these items, the dynamic air bronchogram has been found to be the most effective. However, it is important to note that other LUS items can also offer valuable diagnostic information and in our study with statistical significance. It should also be noted that the presence of dynamic air bronchogram may not always be detectable. Hence, to address this issue, an early diagnosis score has been proposed in this article that takes into consideration all LUS items. Nonetheless, it is important to validate this score through a multicenter study prior to its implementation in clinical practice.
- Conclusion
According to the results of your study, it seems that lung ultrasound was the best tool for diagnosing VAP. Then is it still reasonable to recommend the use of CPIS-PLUS score? Can you provide additional reasoning?
Dear reviewer thank you very much for your statement. After conducting our study, we have determined that the LUS score is the most accurate score to diagnose VAP at an early stage. It is important to note that this study was carried out in a single centre, specifically in a PICU with paediatricians who have a high level of expertise in ultrasound. Despite this limitation, we hold great confidence in our findings. However, to ensure the credibility and validity of our results, we will design and perform an external validation through a multicenter study.
In addition, I would like to draw the attention of the authors to the fact that not all references are consistent with the claims made, and not all claims referring to evidence are supported by references. For example, in the introduction:
Dear reviewer, thank you for your suggestion. We have changed the following references.
The clinical pulmonary infection score (CPIS) has been proposed as a diagnosis tool for VAP in pediatric patients[39,40].
The clinical pulmonary infection score (CPIS) [39,40] has been proposed as a diagnosis tool for VAP in paediatric patients. Page 2 line 77
- There are studies examining the diagnostic accuracy of CPIS in the paediatric population that you do not cite.
We have added these references. Page 2, line 78
- Sachdev, K. Chugh, M. Sethi, D. Gupta, C. Wattal, G. Menon. Clinical Pulmonary Infection Score to Diagnose Ventilator-associated Pneumonia in Children. Indian Pediatr. 2011;48:949-954.
Da Silva PSL, de Aguiar VE, de Carvalho WB, Machado Fonseca MC. Value of clinical pulmonary infection score in critically ill children as a surrogate for diagnosis of ventilator-associated pneumonia. J Crit Care. 2014;29(4):545-550. doi:10.1016/j.jcrc.2014.01.010
Nevertheless, subsequent studies have shown that the CPIS score has a low diagnostic performance, leading to VAP over-diagnose and excessive prescription of antibiotics.
- No reference to specific studies.
We have added specific references. Page 2, line 81.
Marya D. Zilberberg, Andrew F Shorr. Ventilator-Associated Pneumonia: The Clinical Pulmonary Infection Score as a Surrogate for Diagnostics and Outcome. Clinical Infectious Diseases. 2010;51(S1):131-135. doi:10.1086/653062
Shannon M. Fernando, Alexandre Tran, Wei Cheng, et al. Diagnosis of ventilator-associated pneumonia in critically ill adult patients—a systematic review and meta-analysis. Intensive Care Med . 2020;46:1170-1179. doi:https://doi.org/10.1007/s00134-020-06036-z

Reviewer 2 Report
Comments and Suggestions for Authors
Many thanks for the opportunity to review this work entitled: The Clinical Pulmonary Infection Score combined with Procalcitonin and Lung Ultrasound (CPIS-PLUS), a good tool for ventilator associated pneumonia early diagnosis in pediatrics. Authors should specify all the abbreviations in the abstract. In this study LUS resulted with high accuracy in the diagnosis of VAP in children. However, in the real clinical practice, LUS has a lot of limitations as it is an operator-dependent tool and not objective, it doesn’t not provide a panoramic view of both lungs, also for the evaluation of pneumothorax is less sensitive than chest-X ray (Liu, X., Si, S., Guo, Y., & Wu, H. (2022). Limitations of bedside lung ultrasound in neonatal lung diseases. Frontiers in Pediatrics, 10, 855958.). On LUS we can’t have a direct visualization of catheters, tracheal tube and of all complications that can be found in patients with VAP and that can be visualized with a chest x-ray. Therefore this sentence: Lung ultrasound (LUS) is a radiation-free and bedside available alternative that has shown promise in diagnosing VAP in adults[24,25] should be modified. LUS can’t be an alternative of chest –X-ray but an additional tool to use to support the diagnosis of VAP also in adults.
Therefore, the authors should highlight and enhance all the limitations of LUS in the introduction. This sentence in the discussion: But in our study in pediatric population LUS alone was the most useful..should be modified: LUS can’t be used alone or can be the most useful tool because chest-X ray is usually necessary to visualize complications of VAP that can be not visualized on LUS.
Authors should highlight the limits of LUS also in the discussion and I suggest to add some Figures that compare the chest X-ray with LUS.
In the conclusion, should be enhanced the important role of the integrated imaging approach in the management of VAP in children.
Author Response
RESPONSE TO THE REVIEWER COMMENTS
Dear editor and reviewers, thank you very much for giving us the opportunity to improve the manuscript. We hope we have clearly answered all the questions. Please, let us know if further changes are required. All changes in the manuscript are highlighted in yellow.
Reviewers' Comments to Authors
REVIEWER 2
Comments to the Author
Many thanks for the opportunity to review this work entitled: The Clinical Pulmonary Infection Score combined with Procalcitonin and Lung Ultrasound (CPIS-PLUS), a good tool for ventilator associated pneumonia early diagnosis in pediatrics. Authors should specify all the abbreviations in the abstract.
We appreciate the reviewer’s comment and we agree with it. We have added all the abbreviations that were missing in the abstract.
In this study LUS resulted with high accuracy in the diagnosis of VAP in children. However, in the real clinical practice, LUS has a lot of limitations as it is an operator-dependent tool and not objective, it doesn’t not provide a panoramic view of both lungs, also for the evaluation of pneumothorax is less sensitive than chest-X ray (Liu, X., Si, S., Guo, Y., & Wu, H. (2022). Limitations of bedside lung ultrasound in neonatal lung diseases. Frontiers in Pediatrics, 10, 855958.). On LUS we can’t have a direct visualization of catheters, tracheal tube and of all complications that can be found in patients with VAP and that can be visualized with a chest x-ray. Therefore this sentence: Lung ultrasound (LUS) is a radiation-free and bedside available alternative that has shown promise in diagnosing VAP in adults[24,25] should be modified. LUS can’t be an alternative of chest –X-ray but an additional tool to use to support the diagnosis of VAP also in adults.
Therefore, the authors should highlight and enhance all the limitations of LUS in the introduction. This sentence in the discussion: But in our study in pediatric population LUS alone was the most useful should be modified: LUS can’t be used alone or can be the most useful tool because chest-X ray is usually necessary to visualize complications of VAP that can be not visualized on LUS.
Authors should highlight the limits of LUS also in the discussion and I suggest to add some Figures that compare the chest X-ray with LUS.
In the conclusion, should be enhanced the important role of the integrated imaging approach in the management of VAP in children.
Dear reviewer, thank you for your suggestion.
Chest X-ray (CXR) is a commonly used imaging technique for diagnosing various pathologies and checking medical devices, as highlighted in your previous statement, but it can be harmful to patients due to its radiation effects, especially in the case of children, who have many years ahead of them and the cumulative dose of radiation can be harmful. Therefore, it is crucial to weigh the associated risks before performing this procedure.
In our hospital, the care professionals have extensive experience in using ultrasound in different fields. Perhaps for this reason, we may not have emphasized its limitations as much as we should have. Certainly, LUS has some limitations, as it relies on the operator skills and does not provide a panoramic view of both lungs or quantify pneumothorax. In the discussion (page 11, line 387) we highlight some of these limitations in our study: “The present study has several limitations. Firstly, it was conducted in a single centre, in which paediatricians are good skilled in LUS performance, which could potentially affect the reproducibility of these high accuracy results. Secondly, since there is no good gold standard diagnostic test for VAP, it is difficult to establish comparisons”. We have added to the introduction the following sentence (page 2, line 62): However, LUS has limitations as it is operator-dependent and does not provide a panoramic view of both lungs or quantify pneumothorax. A recent study[15] has also found that LUS can diagnose VAP in early stage and with high sensitivity (Sn) and specificity (Sp). Nevertheless, scarce data exists regarding its role and usefulness in pediatric VAP diagnosis, with ongoing investigations[26, 27].
We would like to share some bibliographical references that support the advantages of LUS.
Francesco Raimondi, Nadya Yousef, Fiorella Migliaro, Letizia Capasso, Daniel de Luca. Point-of-care lung ultrasound in neonatology- classification into descriptive and functional applications. Pediatr Res. 2021;90:524-531. doi:https://doi.org/10.1038/s41390-018-0114-9
Hegazy LM, Rezk AR, Sakr HM, Ahmed AS. Comparison of efficacy of lus and cxr in the diagnosis of children presenting with respiratory distress to emergency department. Indian Journal of Critical Care Medicine. 2020;24(6):459-464. doi:10.5005/jp-journals-10071-23459
Saraogi A. Lung ultrasound: Present and future. Lung India. 2015;32(3):250-257. doi:10.4103/0970-2113.156245
Volpicelli G, Elbarbary M, Blaivas M, et al. International evidence-based recommendations for point-of-care lung ultrasound. In: Intensive Care Medicine. Vol 38. ; 2012:577-591. doi:10.1007/s00134-012-2513-4
Bobillo-Perez S, Girona-Alarcon M, Rodriguez-Fanjul J, Jordan I, Balaguer Gargallo M. Lung ultrasound in children: What does it give us? Paediatr Respir Rev. 2020;36:136-141. doi:10.1016/j.prrv.2019.09.006
Volpicelli G, Boero E, Sverzellati N, Cardinale L, Busso M, Boccuzzi F, et al. Semiquantification of pneumothorax volume by lung ultrasound. Intensive Care Med. 2014;40(10):1460–7. 22.
Raimondi F, Rodriguez Fanjul J, Aversa S, Chirico G, Yousef N, De Luca D, et al. Lung Ultrasound for Diagnosing Pneumothorax in the Critically Ill Neonate. J Pediatr. 2015;175:74– 12 78.e1. 23.
Moreno-Aguilar G, Lichtenstein D. Lung ultrasound in the critically ill (LUCI) and the lung point: A sign specific to pneumothorax which cannot be mimicked. Crit Care [Internet]

Round 2
Reviewer 1 Report
Comments and Suggestions for Authors I am generally satisfied with the responses to the comments and the changes to the manuscript that the authors have made. There are still some grammatical errors in the manuscript, but I think they can be edited when preparing the article for printing.Comments on the Quality of English Language There are still some grammatical errors in the manuscript, but I think they can be edited when preparing the article for printing.
Author Response
Dear editor and reviewers, thank you very much for giving us the opportunity to improve the manuscript. We hope we have clearly answered all the questions. Please, let us know if further changes are required. All changes in the manuscript are highlighted in yellow.
Reviewers' Comments to Authors
REVIEWER 1
Comments to the Author
I am generally satisfied with the responses to the comments and the changes to the manuscript that the authors have made. There are still some grammatical errors in the manuscript, but I think they can be edited when preparing the article for printing.
Dear reviewer thank you very much for all your comments that have allowed us to improve the manuscript. We have modified some grammatical errors in the manuscript.

Reviewer 2 Report
Comments and Suggestions for Authors The authors made only some adjustments. However, the limits of LUS are not sufficiently described. As the previous comments, I continue to suggest to extend in the discussion the limits of LUS.Rizvi, Munaza Batool, and Joni E. Rabiner. "Pediatric point-of-care lung ultrasonography: A narrative review." Western Journal of Emergency Medicine 23.4 (2022): 497. Bhalla, Deeksha, et al. "Pediatric lung ultrasonography: Current perspectives." Pediatric Radiology 52.10 (2022): 2038-2050.
I continue to don't visualize Figures with LUS findings of VAP. Authors should also add asome images of chest x Ray findings and LUS if is possible of the same cases (VAP consolidations on chest X ray and on LUS)
Author Response
Dear editor and reviewers, thank you very much for giving us the opportunity to improve the manuscript. We hope we have clearly answered all the questions. Please, let us know if further changes are required. All changes in the manuscript are highlighted in yellow.
Reviewers' Comments to Authors
REVIEWER 2
Comments to the Author
The authors made only some adjustments. However, the limits of LUS are not sufficiently described. As the previous comments, I continue to suggest to extend in the discussion the limits of LUS. Rizvi, Munaza Batool, and Joni E. Rabiner. "Pediatric point-of-care lung ultrasonography: A narrative review." Western Journal of Emergency Medicine 23.4 (2022): 497. Bhalla, Deeksha, et al. "Pediatric lung ultrasonography: Current perspectives." Pediatric Radiology52.10 (2022): 2038-2050.
Dear reviewer, thank you very much for your comment. We apologise if the changes were not clear or specific enough. We have tried to improve and we have added in the discussion lines 322-332 on page 10, the following paragraph:
“LUS can improve early diagnosis of lung diseases in critically ill children, being more sensitive than CXR and comparable to CT, without radiation exposure. Although it is operator-dependent, different studies have shown that even beginners can perform it with high precision, showing adequate interobserver agreement in image interpretation. Obesity and subcutaneous emphysema are the primary patient-related factors that can limit the effectiveness of LUS. However, pediatric patients are usually good candidates for LUS as they have less subcutaneous tissue. There is a limitation to using LUS for the location of pulmonary pathology since it cannot visualize the etiologies that do not extend to the pleural surface. However, more than 95% of pulmonary pathological processes have a pleural component in both adults and paediatric patients, making ultrasound useful in most cases”
I continue to don't visualize Figures with LUS findings of VAP. Authors should also add some images of chest x Ray findings and LUS if is possible of the same cases (VAP consolidations on chest X ray and on LUS)
We have added Figure S1 to the supplementary material to compare CXR and LUS findings in the same patient who was diagnosed with confirmed VAP. We describe LUS findings of VAP in the text and images.
Figure S1. Comparison of CXR and LUS findings in the same patient
This patient has been diagnosed with confirmed VAP. The CXR (image 1) showed a retrocardiac consolidation in the left inferior lung lobe. LUS (image 2) revealed a consolidation with a fragmented pleural line, tissue-like pattern and shred sign (¨) in the posterior segment of the left lung. This consolidation was unilateral, about 40 mm in size and was observed with a dynamic air bronchogram (*), and also fluid bronchogram. Confluent, trailing B-lines were observed arround the consolidation. The application of color Doppler demonstrated vascularization into the consolidation (image 3).
